Subject Areas:
civil engineering

Keywords:
damage index, point load strength, integrality coefficient, P-wave velocity, rock mass

Author for correspondence:
Lei Wen
e-mail: 821149277@qq.com

# A new method for evaluating the rock mass damage index based on the field point load strength

Lei Wen[1], Zhou quan Luo[1], Shi jiao Yang[2],
Ya guang Qin[1], Shao wei Ma[1] and Hong Jiang[1]

[1]School of Resources and Safety Engineering, Central South University, Changsha 410083, Hunan, People's Republic of China
[2]School of Nuclear Resources Engineering, University of South China, Hengyang 421001, People's Republic of China

(iD) LW, 0000-0003-4743-1177

The damage index is a crucial controlling parameter for rock mass strength and deformation in civil, geological and mining engineering projects. In this study, a new method was proposed for computing the damage index of a rock mass based on the field point load strength. This method provides a strength ratio whose numerator is the point load strength (PLS) of splitting along the pre-existing joint and whose denominator is the PLS of intact rock failure. To validate this method, the authors derived a relationship between the damage index and the integrality coefficient using an empirical relation. Moreover, numerous experimental tests were conducted, including P-wave testing and on-site point load testing. Finally, linear regression analysis was performed to analyse the correlation between the new damage index $D_R$ and the integrality coefficient $K_V$. This study demonstrates that the presented new method is sensitive to both plasticity and damage in estimating the damage degree of rock masses in underground excavation and tunnel engineering.

## 1. Introduction

Natural rock mass differs from most other engineering materials, because it is anisotropic and contains discontinuities, such as joint bedding planes, folds, sheared zones and faults, resulting in an internal structure that produces different degrees of damage and destructuration. Moreover, excavation disturbance leads to further damage, resulting in great difficulties in estimating the physical and mechanical properties of a rock mass. Many

**Table 1.** Empirical studies. $\sigma_c$ is the unconfined compressive strength of intact rock, $\sigma_{cm}$ unconfined compressive strength of rock mass, $RMR$ is rock mass rating, $GSI$ geological strength index, $D$ factor indicating the degree of disturbance due to blast damage and stress relaxation.

| author | empirical relation |
| --- | --- |
| Aydan & Dalgic [8] | $\dfrac{\sigma_{cm}}{\sigma_c} = \dfrac{RMR}{RMR + 6(100 - RMR)}$ |
| Yudhbir & Prinzl [9] | $\dfrac{\sigma_{cm}}{\sigma_c} = e^{\frac{7.65(RMR-100)}{100}}$ |
| Laubscher [10] and Singh et al. [11] | $\dfrac{\sigma_{cm}}{\sigma_c} = \dfrac{RMR - \text{rating for } \sigma_c}{106}$ |
| Ramamurthy et al. [12] and Ramamurthy [13] | $\dfrac{\sigma_{cm}}{\sigma_c} = e^{\frac{RMR-100}{18.75}}$ |
| Kalamaras & Bieniawski [14] | $\dfrac{\sigma_{cm}}{\sigma_c} = e^{\frac{RMR-100}{24}}$ |
| Sheorey [15] | $\dfrac{\sigma_{cm}}{\sigma_c} = e^{\frac{RMR-100}{20}}$ |
| Hoek et al. [16] | $\dfrac{\sigma_{cm}}{\sigma_c} = e^{GSI-100/9-3D\left[\frac{1}{2}+\frac{1}{6}\left(e^{-\frac{GSI}{15}}-e^{\frac{20}{3}}\right)\right]}$ |
| AASHTO [17] | $\dfrac{\sigma_{cm}}{\sigma_c} = 0.0231RQD - 1.32 \geq 0.15$ |
| Lian-yang Zhang [18] | $\dfrac{\sigma_{cm}}{\sigma_c} = 10^{0.013RQD-1.34}$ |

methods for assessing rock mass properties have been proposed and used in engineering practice, such as the rock quality designation ($RQD$) [1], the rock mass rating ($RMR$) [2,3], the Rock Mass Quality Classification Q-system (Q) [4,5] and the geological strength index ($GSI$) [6,7]. Additionally, several researchers have extensively studied the uniaxial compressive strength ratio $\sigma_{cm}/\sigma_c$ of joint rock mass, and regression analysis has been used to analyse the correlation between the ratio $\sigma_{cm}/\sigma_c$ and $RMR$ or $GSI$, as shown in table 1, where $\sigma_c$ is the unconfined compressive strength of intact rock and $\sigma_{cm}$ is the unconfined compressive strength of a rock mass. Kulhawy & Goodman [19] and AASHTO [17] reported the variation of the ratio $\sigma_{cm}/\sigma_c$ with $RQD$. Lian-yang Zhang [18] reported the ratio $\sigma_{cm}/\sigma_c$ versus $RQD$ relation for estimating the strength of a jointed rock mass. Several researchers [8,10,12–15,20] reported the ratio $\sigma_{cm}/\sigma_c$ versus $RMR$ relation for estimating the uniaxial compressive strength of a jointed rock mass. Hoek et al. [16] reported the relationships of the ratio $\sigma_{cm}/\sigma_c$ with $RMR$, $GSI$ and $D$, where $D$ is the degree of disturbance due to blast damage and stress relaxation. These empirical relations provide a reliable basis for the evaluation of rock mass strength. In practice, the engineering properties of rock mass are nearly impossible to measure directly. Thus, many researchers have described a damaged rock mass based on classification indices, such as $GSI$ [7,21–23], $RQD$ [18,24–26] and $RMR$ [8,9,15,24,27–29].

According to the above-mentioned studies, researchers have focused on the determination of the ratio $\sigma_{cm}/\sigma_c$, $RMR$, $GSI$ and $RQD$, with the methods used varying among researchers and rock mass types. It is a little difficult to measure the value of $\sigma_{cm}$ of rock mass (or damaged rock) directly; the ratio $\sigma_{cm}/\sigma_c$ can only be obtained using an empirical relation, as reported by Ramamurthy [30], Singh et al. [11] and Singh & Rao [31]. The determination of $RMR$ parameters is also relatively complex; these parameters include rock strength, $RQD$ value, joint space, joint condition and groundwater. $GSI$ characterizes the degree to which the strength of a rock mass is weakened under various geological conditions, and is used to describe the characteristics of rock masses in detail. The $GSI$ parameter is essentially a qualitative parameter that includes six factors: joints distribution, block shape, the degree of geological disturbance, joint roughness, the weathering degree of the joints and filling situation. The determination of the $RQD$ value depends mainly on the drilling technology and mechanical equipment used. In research, the uniaxial compressive strength ratio $\sigma_{cm}/\sigma_c$, $RMR$, $GSI$ and $RQD$ are

the parameters available for describing rock mass properties. However, the method for determining these indices is relatively complicated and tedious, and some indices are only qualitatively estimated based on experience. Thus, in this paper, a new method is proposed for determining the rock mass damage index based on field point load test results.

Field point load tests are preferred because they have strong applicability, as they are flexible in simple testing. Hence, this paper focuses on the determination of the field point load strength ratio and its utilization for evaluating the engineering properties (mainly damage index and integrality coefficient) of rock masses. This new damage index is a point load strength ratio whose numerator is the point load strength (PLS) of splitting along the pre-existing joint and whose denominator is the PLS of the failure of intact rock. One of the critical objectives is to find samples of splitting along a pre-existing joint in terms of significant joint properties of a failure surface. Next, a series of theoretical derivations of the relationship between the field point load strength ratio and the integrality coefficient are presented and briefly discussed. Finally, a new method for calculating the damage index is proposed to determine the damage index of a rock mass. The index can be used for estimating the elastic modulus and unconfined compressive strength of rock masses. To validate this method, various experimental designs and analyses are provided in §§3–5; the discussion and conclusion are presented in §§6 and 7, respectively. This paper outlines the key aspects involved in determining the field point load strength ratio and provides useful information for effectively calculating the integrality coefficient or damage index of rock masses.

# 2. Field point load strength

The point load strength test has been regarded as an inexpensive and effective testing method for the estimation of the strength of rocks because of its ease of testing, simplicity of specimen preparation and potential field applications [32–35]. The test has been referred to as an indirect method for assessing the tensile or compressive strength of rocks [36]. Some researchers have attempted to establish empirical relations between the uniaxial compressive strength (UTS)/Brazilian tensile strength (BTS) and the point load strength in applying the point load test to various rock types [32,33,37]. In this study, point load strength tests were performed on irregular rock blocks using a digital point load test system according to the guidelines of the ASTM [38]. The study site was the Huize lead and zinc mine, straddling the provinces of YunNan and GuiZhou in southwest China, as shown in figure 1.

Four types of samples of dolomite and limestone were adopted in this test; these four samples originate from different stratigraphic locations, as shown in figure 2. Study area is composed of the northeast to southwest fold and fault, forming the anticline. The sampling location is located between the Qi-lin Chang fault and the Yin-chang Po fault. All samples belong to Palaeozoic strata. The four types of rock are referred to as $C_{1d}$, $C_{1b}$, $C_{2w}$ and $P_{1q+m}$. $C_{1d}$, $C_{1b}$ and $C_{2w}$ belong to the Carboniferous system. $P_{1q+m}$ belongs to the Permian system. $C_{1d}$ is dark-grey and grey cryptocrystalline limestone. $C_{1b}$ is a shallow, pale and flesh-pink coarse-grain dolomite. $C_{2w}$ is light-grey to dark-grey limestone and dolomitic limestone. $P_{1q+m}$ is dark-grey to light-grey, thin-mouth to cryptocrystalline limestone and dolomitic limestone. The physical and mechanical parameters of all rock samples employed are presented in table 2, where $V_p$ is the P-wave velocity of each rock sample.

The point load tester used in this study consisted of a small hydraulic pump, a hydraulic jack, a pressure gauge and an interchangeable testing frame with a very high transverse stiffness. Panek *et al*. [39] elaborate the detailed process of point load test. In this paper, the details of the technique of field test are carried out as shown in figure 3. The thickness of the irregular rock blocks ranged from 30 to 70 mm, and their lengths were less than 150 mm, as shown in figure 3. All of the point load tests were performed on irregular rock blocks from four different types of sedimentary rock. Each irregular rock block was slowly loaded until failure. According to the failure mode, the measured results could be divided into two failure types based on visual inspection alone: intact rock failure in which no apparent joint phenomenon could be observed on the failure surface of the samples, and splitting along a pre-existing joint. When it was not clear whether a joint occurred on a sample failure surface, the sample was excluded. Thus, a total of 193 irregular block samples were obtained. The raw data of all specimens are presented in table 3, where $L$, $W$, $H$ are the maximum length, width and height of the irregular samples before testing, respectively; $P$ is the maximum load applied during loading; $W_f$ is the effective width of the fracture surface; and $H_D$ is the distance between the loading points; the letter 'I' represents samples exhibiting intact rock failure; and the letter 'J' represents samples exhibiting splitting along a pre-existing joint.

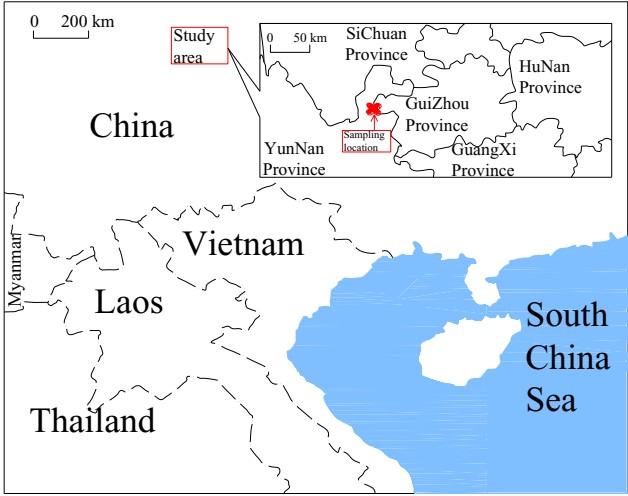

**Figure 1.** Study area.

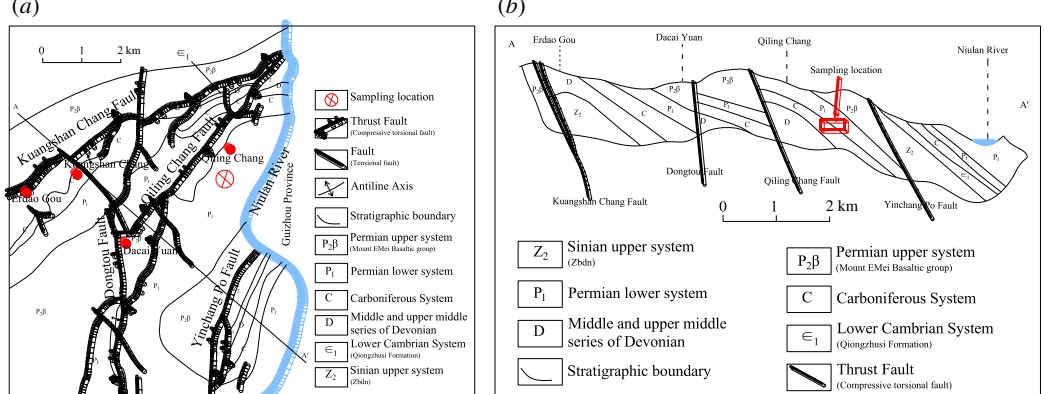

**Figure 2.** Stratigraphic and geological structure.

Irregular rock blocks must be corrected to the standard equivalent diameter ($D_e$) of 50 mm. Size correction can be performed graphically and mathematically, as suggested by ASTM [38] procedures. The point load index PLS is determined by the following equations:

$$D_e = \sqrt{\frac{4 H_D W_f}{\pi}} \tag{2.1}$$

and

$$\mathrm{PLS} = \frac{4P}{\pi D_e^2}, \tag{2.2}$$

where $P$ is the failure load and $D_e$ is the equivalent diameter of irregular blocks; $H_D$ and $W_f$ are of the maximum length and average width of the failure surface in millimetres, respectively (figure 3). As shown in table 3, the fluctuation range of PLS value is larger. The main reason for the large fluctuation range of PLS value is the irregularity of rock sample and the different degree of micro-cracks in the sample. According to the results shown in table 3 and table 4, the PLS value of $C_{1d}$ fluctuated between 1.22 and 7.9 MPa, with mean values of 2.36 MPa in cases of splitting along a pre-existing joint and 4.41 MPa in cases of intact rock failure. The PLS value of $C_{1b}$ was found to vary in a broad range (0.63–7.93 MPa), with a mean value of 2.56 MPa, a mean value of 2.14 MPa for splitting along a pre-existing joint and a mean value of 4.38 MPa for intact rock failure. The PLS values of $C_{2w}$ were found to be in the range 0.66–7.48 MPa, with a mean value of 3.44 MPa, a mean value of 2.39 MPa for splitting along a pre-existing joint and a mean value of 4.49 MPa for intact rock failure. The PLS value of $P_{1q+m}$ fluctuated between 2.1 and 8.9 MPa, with a mean value of 4.22 MPa, a mean value of 2.46 MPa for splitting along a pre-existing joint and a mean value of 5.21 MPa for intact rock failure.

**Table 2.** Physical and mechanical parameters of rock samples.

| no. | geological age | rock type | test times | elastic modulus (GPa) | density (t m$^{-3}$) | $V_p$ (m s$^{-1}$) | Poisson's ratio | UCS (MPa) | | | BTS (MPa) | | |
|---|---|---|---|---|---|---|---|---|---|---|---|---|---|
| | | | | | | | | mean | max. | min. | mean | max. | min. |
| C$_{1d}$ | Carboniferous | limestone | 10 | 30.4 | 2.70 | 5344 | 0.28 | 69.9 | 88.34 | 56.12 | 4.87 | 7.879 | 3.721 |
| C$_{1b}$ | Carboniferous | dolomite | 10 | 25.13 | 2.72 | 4744 | 0.25 | 59.52 | 75.85 | 44.86 | 4.08 | 5.243 | 2.991 |
| C$_{2w}$ | Carboniferous | limestone | 10 | 29.98 | 2.71 | 5130 | 0.24 | 75.95 | 124.74 | 32.13 | 5.43 | 6.53 | 3.873 |
| P$_{1q+m}$ | Permian | limestone | 10 | 21.13 | 2.73 | 5514 | 0.25 | 60.69 | 86.62 | 49.34 | 4.44 | 5.653 | 2.797 |

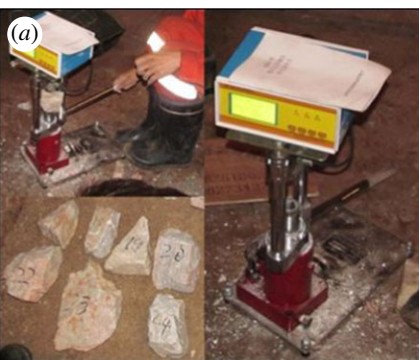
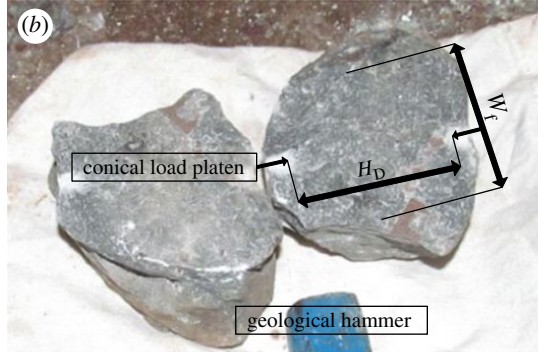

**Figure 3.** Point load test process: test process, samples and tester (*a*) and sample failure surface (*b*).

In previous studies, samples with splitting along a pre-existing joint were excluded from the set of tested samples [37,40]. In this study, the results of all point load tests were recorded clearly. Based on all of the test results collected, a new conclusion is drawn. The sum of the ratio $\mathrm{PLS_{sm}}/\mathrm{PLS_s}$ and the integrality coefficient $K_V$ is approximately equal to 1, where $\mathrm{PLS_{sm}}$ is the PLS of rock samples with splitting along a pre-existing joint, $\mathrm{PLS_s}$ is the PLS of samples with intact rock failure and $K_V$ is the integrality coefficient. The integrality coefficient of a rock mass is determined as suggested by GB50218–94 [41] in China and is expressed as follows:

$$K_V = \frac{V_P'^2}{V_p^2},$$  (2.3)

where $V_P$ is the P-wave velocity of intact rock and $V_p'$ is the P-wave velocity of a rock mass.

The new damage index $D_R$ can be defined as the ratio of the value for splitting along a pre-existing joint $I_{sm}$ to the value for intact rock failure $\mathrm{PLS_s}$ (this sentence explains equation (2.4)). The new damage index $D_R$ is expressed as follows:

$$D_R = \frac{\mathrm{PLS_{sm}}}{\mathrm{PLS_s}},$$  (2.4)

where $\mathrm{PLS_{sm}}$ is the PLS of rock samples with splitting along a pre-existing joint, $\mathrm{PLS_s}$ is the PLS of samples with intact rock failure.

The field point load strength ratio is presented in table 5. These results show that the integrality coefficient $K_V$ is approximately equal to the difference $1 - D_R$, and the deviation between $K_V$ and the difference $1 - D_R$ is also listed in table 5. To verify this conclusion, a theoretical derivation is conducted in §3 and the rock mass joint statistics of the study area and the field experimental tests are derived in §4.

# 3. Theoretical derivation of integrity and damage index

## 3.1. Damage index and elastic modulus

Because a rock mass is a complex and heterogeneous material, its damage level cannot be directly measured by current methods. In other words, it is difficult to measure the damage area of damaged rock. However, the strength of damaged rock can be accurately measured in a mechanics laboratory or by *in situ* tests. In this respect, a new method for calculating the damage index of a rock mass is proposed via *in situ* point load strength testing, as formulated in equation (2.4).

According to Lemaitre [42], the definition of the damage variable $D$ is different from the new damage index $D_R$ in equation (2.4). It is assumed that the newly defined $D_R$ is reasonable and it can replace the damage variable $D$ of Lemaitre's [42] definition. Like that, the newly defined $D_R$ is tried to derive a series of theoretical relationships. Finally, the assumption is verified by field point load strength test.

**Table 3.** The test parameters of point load strength. L, W, H are the maximum length, width and height of failure surface, the letter 'I' represents samples exhibiting intact rock failure; and the letter 'J' represents samples exhibiting splitting along a pre-existing joint, P is the maximum load applied during loading; $W_f$ is the effective width of the fracture surface; and $H_D$ is the distance between the loading points.

| no. | size (mm) | | | P (kN) | $H_D$ (mm) | $W_f$ (mm) | PLS | failure modes |
|---|---|---|---|---|---|---|---|---|
| | L | W | H | | | | | |
| C1d is dark grey and grey cryptocrystalline limestone | | | | | | | | |
| 1 | 95 | 61 | 41 | 16.17 | 36 | 75 | 4.70 | I |
| 2 | 80 | 64 | 38 | 13.52 | 34 | 55 | 5.67 | I |
| 3 | 116 | 61 | 48 | 22.98 | 35 | 94 | 5.48 | I |
| 4 | 125 | 75 | 36 | 13.47 | 38 | 72 | 3.86 | I |
| 5 | 81 | 63 | 35 | 38.48 | 33 | 75 | 1.22 | J |
| 6 | 115 | 65 | 40 | 8.88 | 39 | 76 | 2.35 | J |
| 7 | 71 | 59 | 59 | 20.59 | 58 | 70 | 3.98 | J |
| 8 | 107 | 86 | 47 | 24.14 | 41 | 83 | 5.57 | I |
| 9 | 120 | 73 | 43 | 28.80 | 45 | 74 | 2.79 | J |
| 10 | 89 | 65 | 30 | 8.71 | 31 | 79 | 6.78 | I |
| 11 | 79 | 46 | 33 | 10.03 | 31 | 40 | 6.35 | I |
| 12 | 95 | 69 | 31 | 9.44 | 31 | 69 | 3.47 | I |
| 13 | 96 | 65 | 31 | 10.97 | 36 | 88 | 2.72 | I |
| 14 | 140 | 85 | 45 | 16.98 | 48 | 141 | 1.97 | I |
| 15 | 101 | 84 | 65 | 10.16 | 60 | 92 | 1.45 | J |
| 16 | 95 | 70 | 37 | 16.49 | 39 | 98 | 3.39 | I |
| 17 | 100 | 80 | 38 | 8.86 | 29 | 66 | 3.63 | I |
| 18 | 103 | 74 | 36 | 18.79 | 35 | 94 | 4.48 | I |
| 19 | 101 | 65 | 37 | 16.71 | 42 | 92 | 3.40 | I |
| 20 | 113 | 61 | 57 | 15.52 | 58 | 72 | 4.37 | J |
| 21 | 105 | 91 | 46 | 22.21 | 42 | 95 | 4.37 | I |
| 22 | 90 | 53 | 40 | 21.45 | 40 | 61 | 6.90 | I |
| 23 | 74 | 53 | 39 | 14.93 | 38 | 51 | 6.05 | I |
| 24 | 110 | 79 | 38 | 3.98 | 43 | 58 | 1.25 | J |
| 25 | 105 | 46 | 34 | 9.06 | 34 | 45 | 4.65 | J |
| 26 | 135 | 93 | 69 | 24.49 | 65 | 115 | 2.57 | I |
| 27 | 130 | 82 | 30 | 5.22 | 30 | 92 | 1.48 | J |
| 28 | 140 | 49 | 34 | 11.18 | 30 | 75 | 3.90 | I |
| 29 | 135 | 81 | 52 | 17.27 | 58 | 86 | 2.72 | J |
| 30 | 106 | 87 | 58 | 22.94 | 50 | 112 | 3.22 | I |
| 31 | 110 | 80 | 41 | 13.91 | 32 | 110 | 3.10 | J |
| 32 | 67 | 65 | 33 | 13.60 | 29 | 49 | 7.51 | I |
| 33 | 76 | 61 | 45 | 25.52 | 39 | 65 | 7.90 | I |
| 34 | 115 | 65 | 32 | 5.10 | 29 | 62 | 2.23 | J |
| 35 | 78 | 55 | 47 | 20.72 | 34 | 71 | 6.74 | I |
| 36 | 105 | 90 | 37 | 9.37 | 29 | 77 | 3.29 | J |
| 37 | 77 | 70 | 45 | 14.23 | 40 | 64 | 4.36 | I |

(Continued.)

| no. | size (mm) L | W | H | $P$ (kN) | $H_D$ (mm) | $W_f$ (mm) | PLS | failure modes |
|---|---|---|---|---|---|---|---|---|
| 38 | 76 | 60 | 50 | 17.30 | 29 | 81 | 5.78 | I |
| 39 | 130 | 75 | 56 | 20.64 | 54 | 105 | 2.86 | I |
| 40 | 90 | 63 | 45 | 19.60 | 36 | 72 | 5.77 | I |
| 41 | 133 | 96 | 40 | 10.92 | 39 | 133 | 1.65 | J |
| 42 | 129 | 91 | 44 | 17.86 | 49 | 87 | 3.29 | I |
| 43 | 165 | 92 | 39 | 7.95 | 37 | 85 | 1.98 | J |
| 44 | 136 | 66 | 52 | 21.66 | 48 | 62 | 5.71 | I |
| 45 | 165 | 79 | 43 | 15.76 | 30 | 96 | 4.29 | I |
| 46 | 128 | 84 | 34 | 8.98 | 38 | 87 | 2.13 | J |
| 47 | 123 | 92 | 42 | 17.73 | 45 | 110 | 2.81 | I |
| 48 | 129 | 99 | 46 | 21.18 | 38 | 93 | 4.70 | I |
| 49 | 118 | 89 | 51 | 22.21 | 59 | 89 | 3.32 | I |
| 50 | 145 | 82 | 61 | 20.40 | 60 | 115 | 2.32 | I |
| 52 | 104 | 88 | 42 | 12.56 | 40 | 88 | 2.80 | I |
| 53 | 136 | 94 | 45 | 15.74 | 47 | 96 | 2.74 | I |
| 54 | 91 | 68 | 53 | 16.28 | 54 | 88 | 2.69 | I |
| 55 | 128 | 89 | 51 | 12.67 | 51 | 106 | 1.84 | J |
| 56 | 116 | 83 | 53 | 22.82 | 51 | 118 | 2.98 | I |
| 57 | 110 | 86 | 42 | 26.12 | 37 | 106 | 5.23 | I |
| 58 | 145 | 96 | 34 | 20.51 | 29 | 132 | 4.21 | I |
| $C_{2w}$ is light grey to dark grey limestone and dolomitic limestone | | | | | | | | |
| 1 | 90 | 75 | 39 | 17.32 | 39 | 91 | 3.83 | I |
| 2 | 96 | 84 | 49 | 12.35 | 55 | 61 | 2.89 | J |
| 3 | 146 | 100 | 49 | 24.87 | 49 | 99 | 4.02 | I |
| 4 | 98 | 79 | 34 | 35.54 | 38 | 101 | 7.27 | I |
| 5 | 104 | 97 | 35 | 16.12 | 39 | 98 | 3.31 | I |
| 6 | 92 | 87 | 48 | 12.23 | 49 | 75 | 2.61 | J |
| 7 | 145 | 91 | 34 | 11.97 | 33 | 105 | 2.71 | J |
| 8 | 112 | 67 | 36 | 20.42 | 38 | 71 | 5.94 | I |
| 9 | 105 | 92 | 35 | 19.96 | 41 | 104 | 3.67 | I |
| 10 | 137 | 101 | 45 | 41.85 | 41 | 90 | 8.90 | I |
| 11 | 125 | 79 | 34 | 17.19 | 34 | 61 | 6.51 | I |
| 12 | 114 | 109 | 49 | 25.77 | 56 | 75 | 4.82 | I |
| 13 | 114 | 89 | 49 | 17.92 | 49 | 104 | 2.76 | J |
| 14 | 133 | 90 | 57 | 13.50 | 50 | 101 | 2.10 | J |
| 15 | 114 | 84 | 52 | 15.92 | 51 | 107 | 2.29 | J |
| 16 | 131 | 93 | 59 | 24.93 | 55 | 92 | 3.87 | I |
| 17 | 135 | 97 | 57 | 35.12 | 58 | 114 | 4.17 | J |
| 18 | 95 | 56 | 35 | 2.53 | 39 | 30 | 1.7 | J |
| 19 | 125 | 93 | 40 | 11.02 | 39 | 95 | 2.33 | J |

| no. | size (mm) | | | *P* (kN) | $H_D$ (mm) | $W_f$ (mm) | *PLS* | failure modes |
| --- | --- | --- | --- | --- | --- | --- | --- | --- |
| | *L* | *W* | *H* | | | | | |
| P$_{1q+m}$ is dark grey to light grey cryptocrystalline limestone | | | | | | | | |
| 1 | 123 | 83 | 35 | 8.41 | 36 | 68 | 2.70 | J |
| 2 | 134 | 94 | 49 | 21.83 | 38 | 80 | 5.64 | I |
| 3 | 98 | 75 | 43 | 11.56 | 42 | 71 | 3.04 | I |
| 4 | 120 | 67 | 31 | 11.56 | 25 | 82 | 4.43 | I |
| 5 | 134 | 84 | 45 | 20.13 | 44 | 79 | 4.55 | I |
| 6 | 135 | 93 | 63 | 29.34 | 59 | 88 | 4.44 | I |
| 7 | 120 | 98 | 42 | 16.00 | 50 | 97 | 2.59 | I |
| 8 | 105 | 82 | 51 | 20.86 | 44 | 57 | 6.53 | I |
| 9 | 121 | 56 | 30 | 11.35 | 27 | 79 | 4.18 | I |
| 10 | 77 | 55 | 35 | 15.30 | 23 | 94 | 5.64 | I |
| 11 | 99 | 94 | 34 | 10.35 | 38 | 94 | 2.27 | J |
| 12 | 120 | 94 | 33 | 5.74 | 34 | 115 | 1.15 | J |
| 13 | 94 | 67 | 46 | 12.61 | 38 | 64 | 4.07 | I |
| 14 | 109 | 89 | 53 | 20.37 | 53 | 91 | 3.32 | I |
| 15 | 108 | 93 | 34 | 13.13 | 34 | 86 | 3.53 | J |
| 16 | 125 | 85 | 43 | 20.08 | 39 | 99 | 4.08 | I |
| 17 | 135 | 80 | 44 | 16.33 | 38 | 79 | 4.27 | I |
| 18 | 85 | 79 | 44 | 11.23 | 33 | 75 | 3.56 | I |
| 19 | 86 | 61 | 37 | 12.27 | 33 | 66 | 4.42 | I |
| 20 | 95 | 50 | 32 | 15.25 | 32 | 50 | 7.48 | I |
| 21 | 125 | 95 | 51 | 23.26 | 47 | 115 | 3.38 | I |
| 22 | 130 | 98 | 51 | 13.28 | 49 | 92 | 2.31 | J |
| 23 | 105 | 94 | 38 | 13.44 | 38 | 90 | 3.08 | I |
| 24 | 121 | 62 | 46 | 3.50 | 41 | 102 | 0.66 | J |
| 25 | 131 | 89 | 40 | 10.21 | 31 | 64 | 4.04 | J |
| 26 | 115 | 79 | 42 | 15.82 | 30 | 78 | 5.31 | I |
| 27 | 123 | 90 | 41 | 14.10 | 36 | 84 | 3.66 | I |
| 28 | 130 | 59 | 41 | 17.78 | 45 | 63 | 4.92 | I |
| 29 | 120 | 95 | 46 | 11.80 | 40 | 93 | 2.49 | J |
| 30 | 100 | 86 | 43 | 5.71 | 29 | 72 | 2.15 | J |
| 31 | 135 | 103 | 52 | 24.45 | 42 | 103 | 4.44 | I |
| 32 | 98 | 69 | 33 | 5.79 | 33 | 67 | 2.05 | I |
| 33 | 125 | 76 | 41 | 28.90 | 42 | 94 | 5.74 | I |
| 34 | 105 | 42 | 47 | 22.56 | 38 | 65 | 7.17 | I |
| 35 | 105 | 72 | 38 | 11.16 | 30 | 97 | 3.01 | I |
| 36 | 95 | 60 | 42 | 15.28 | 32 | 81 | 4.63 | I |
| 37 | 150 | 54 | 35 | 11.34 | 36 | 53 | 4.66 | I |
| 38 | 125 | 65 | 37 | 4.31 | 28 | 54 | 2.24 | J |
| 39 | 98 | 65 | 50 | 8.17 | 31 | 93 | 2.23 | I |

| | size (mm) | | | | | | | failure |
| no. | L | W | H | P (kN) | $H_D$ (mm) | $W_f$ (mm) | PLS | modes |
|---|---|---|---|---|---|---|---|---|
| 40 | 88 | 67 | 50 | 22.78 | 48 | 85 | 4.38 | I |
| 41 | 130 | 85 | 45 | 5.34 | 42 | 85 | 1.17 | J |
| 42 | 77 | 65 | 47 | 13.21 | 39 | 76 | 3.50 | I |
| 43 | 125 | 85 | 30 | 6.81 | 24 | 42 | 5.30 | J |
| 44 | 105 | 64 | 43 | 13.74 | 46 | 57 | 4.11 | I |
| 45 | 100 | 65 | 57 | 31.61 | 53 | 70 | 6.69 | I |
| 46 | 75 | 64 | 50 | 21.78 | 42 | 82 | 4.97 | I |
| 47 | 94 | 65 | 28 | 14.98 | 25 | 81 | 5.81 | I |
| 48 | 155 | 95 | 37 | 21.50 | 24 | 115 | 6.11 | I |
| 49 | 110 | 75 | 50 | 5.23 | 43 | 53 | 1.81 | J |
| 50 | 120 | 97 | 50 | 8.43 | 40 | 98 | 1.69 | J |
| 51 | 143 | 90 | 55 | 37.14 | 48 | 100 | 6.07 | I |
| 52 | 95 | 48 | 45 | 9.13 | 44 | 67 | 2.43 | I |
| $C_{1b}$ is shallow pale and flesh pink coarse-grain dolomite | | | | | | | | |
| 1 | 77 | 57 | 45 | 10.14 | 40 | 91 | 2.19 | I |
| 2 | 76 | 73 | 38 | 13.83 | 33 | 53 | 6.21 | I |
| 3 | 79 | 69 | 42 | 13.80 | 43 | 82 | 3.07 | I |
| 4 | 122 | 48 | 38 | 3.63 | 32 | 79 | 1.13 | J |
| 5 | 61 | 46 | 39 | 2.77 | 37 | 62 | 0.95 | J |
| 6 | 110 | 86 | 39 | 6.93 | 39 | 86 | 1.62 | J |
| 7 | 96 | 62 | 42 | 12.15 | 42 | 76 | 2.99 | J |
| 8 | 89 | 64 | 37 | 6.34 | 40 | 61 | 2.04 | I |
| 9 | 112 | 75 | 54 | 16.97 | 46 | 71 | 4.08 | I |
| 10 | 77 | 62 | 35 | 21.02 | 32 | 65 | 7.93 | I |
| 11 | 85 | 65 | 33 | 14.50 | 32 | 42 | 8.47 | I |
| 12 | 93 | 75 | 38 | 1.98 | 35 | 51 | 3.51 | J |
| 13 | 91 | 57 | 42 | 11.72 | 42 | 65 | 3.37 | I |
| 14 | 105 | 49 | 32 | 5.02 | 32 | 65 | 1.90 | J |
| 15 | 87 | 69 | 36 | 9.59 | 39 | 85 | 2.27 | J |
| 16 | 113 | 81 | 43 | 4.04 | 44 | 115 | 0.63 | J |
| 17 | 105 | 79 | 34 | 5.79 | 34 | 95 | 1.41 | J |
| 18 | 109 | 42 | 34 | 4.04 | 32 | 101 | 0.98 | J |
| 19 | 75 | 65 | 45 | 9.81 | 52 | 74 | 2.00 | J |
| 20 | 115 | 46 | 34 | 5.72 | 32 | 52 | 2.70 | I |
| 21 | 116 | 73 | 32 | 7.08 | 31 | 94 | 1.91 | J |
| 22 | 89 | 73 | 45 | 22.34 | 34 | 67 | 7.70 | I |
| 23 | 81 | 46 | 44 | 6.79 | 44 | 49 | 2.47 | I |
| 24 | 95 | 49 | 39 | 1.99 | 38 | 102 | 0.40 | J |
| 25 | 109 | 77 | 33 | 15.63 | 40 | 95 | 3.23 | I |
| 26 | 97 | 74 | 31 | 14.09 | 38 | 105 | 2.77 | I |

(*Continued.*)

| no. | size (mm) | | | P (kN) | $H_D$ (mm) | $W_f$ (mm) | PLS | failure modes |
|---|---|---|---|---|---|---|---|---|
| | L | W | H | | | | | |
| 27 | 104 | 98 | 47 | 30.73 | 47 | 96 | 5.35 | I |
| 28 | 117 | 76 | 39 | 11.24 | 41 | 75 | 2.87 | J |
| 29 | 82 | 76 | 33 | 10.68 | 36 | 73 | 3.19 | J |
| 30 | 115 | 75 | 33 | 20.48 | 33 | 65 | 7.49 | I |
| 31 | 92 | 86 | 44 | 15.76 | 41 | 59 | 5.11 | I |
| 32 | 105 | 76 | 46 | 15.96 | 43 | 97 | 3.00 | I |
| 33 | 128 | 48 | 33 | 13.74 | 33 | 65 | 5.03 | I |
| 34 | 94 | 75 | 43 | 7.23 | 46 | 89 | 1.39 | J |
| 35 | 68 | 47 | 30 | 8.65 | 25 | 47 | 5.78 | I |
| 36 | 98 | 72 | 59 | 28.87 | 58 | 59 | 6.62 | I |
| 37 | 91 | 48 | 45 | 18.69 | 42 | 51 | 6.85 | I |
| 38 | 135 | 80 | 40 | 18.84 | 40 | 110 | 3.36 | I |
| 39 | 120 | 85 | 50 | 6.52 | 50 | 75 | 1.36 | I |
| 40 | 125 | 80 | 37 | 8.95 | 40 | 120 | 1.46 | I |
| 41 | 135 | 85 | 60 | 15.49 | 55 | 125 | 1.77 | J |
| 42 | 135 | 85 | 55 | 14.15 | 40 | 50 | 5.55 | I |
| 43 | 110 | 75 | 37 | 9.73 | 35 | 120 | 1.82 | J |
| 44 | 100 | 90 | 30 | 11.97 | 30 | 90 | 3.48 | I |
| 45 | 100 | 85 | 60 | 20.84 | 55 | 85 | 3.50 | I |
| 46 | 105 | 67 | 38 | 6.65 | 40 | 35 | 3.73 | J |
| 47 | 115 | 85 | 45 | 11.88 | 40 | 85 | 2.74 | J |
| 48 | 100 | 65 | 38 | 13.98 | 40 | 67 | 4.09 | I |
| 49 | 130 | 95 | 58 | 9.71 | 50 | 88 | 1.73 | J |
| 50 | 108 | 10 | 47 | 7.60 | 50 | 100 | 1.19 | J |
| 51 | 90 | 75 | 33 | 5.71 | 35 | 45 | 2.84 | J |
| 52 | 100 | 55 | 43 | 12.30 | 40 | 64 | 3.77 | I |
| 53 | 110 | 78 | 48 | 8.00 | 40 | 105 | 1.49 | J |
| 54 | 105 | 76 | 63 | 27.51 | 65 | 58 | 5.73 | J |
| 55 | 107 | 60 | 45 | 16.62 | 40 | 94 | 3.47 | J |
| 56 | 108 | 55 | 46 | 10.35 | 51 | 60 | 2.66 | I |
| 57 | 100 | 85 | 46 | 13.99 | 50 | 100 | 2.20 | J |
| 58 | 96 | 85 | 44 | 13.99 | 34 | 96 | 3.37 | I |
| 59 | 139 | 50 | 50 | 16.41 | 42 | 48 | 6.39 | I |
| 60 | 103 | 39 | 25 | 11.73 | 27 | 82 | 4.16 | I |

Based on the stress–strain relationship of intact rock under uniaxial compressive testing, the elastic strain $\varepsilon_r$ of intact rock is represented as follows:

$$\varepsilon_r = \frac{\sigma_c}{E},$$  (3.1)

where $\sigma_c$ and $E$ are the uniaxial compressive strength (UCS) and the elastic modulus of the intact rock, respectively.

**Table 4.** Average values of PLS on different rock types.

| rock type | $C_{1d}$ | | | $C_{1b}$ | | | $C_{2w}$ | | | $P_{1q+m}$ | | |
|---|---|---|---|---|---|---|---|---|---|---|---|---|
| | limestone | | | dolomite | | | limestone | | | limestone | | |
| | J | l | all | J | l | all | J | l | all | J | l | all |
| number | 18 | 40 | 58 | 27 | 33 | 60 | 14 | 38 | 52 | 7 | 10 | 17 |
| mean | 2.36 | 4.41 | 3.75 | 2.14 | 4.38 | 2.56 | 2.39 | 4.49 | 3.44 | 2.46 | 5.21 | 4.22 |
| s.d. | 2.19 | 1.356 | 1.665 | 1.32 | 1.938 | 1.97 | 1.818 | 1.341 | 1.598 | 0.59 | 1.91 | 1.85 |

**Table 5.** Calculated $D_R$ value based on field point load strength.

| name | $C_{1d}$ | $C_{1b}$ | $C_{2w}$ | $P_{1q+m}$ |
|---|---|---|---|---|
| $D_R$ | 0.535 | 0.489 | 0.532 | 0.472 |
| $1 - D_R$ | 0.465 | 0.511 | 0.468 | 0.528 |
| $K_V$ | 0.453 | 0.496 | 0.481 | 0.517 |
| deviation between $K_V$ and $1 - D_R$ | 2.24% | 3.1% | 2.4% | 2.3% |

According to Lemaitre [42], the relationship between the elastic modulus $E$ of intact rock and the elastic modulus $E_m$ of a rock mass can be expressed as follows:

$$E_m = \frac{E}{1 - D_R}. \tag{3.2}$$

Based on the equivalent stress proposed by Lemaitre [42], it is assumed that the deformation of a rock mass can be represented by the equivalent stress. The elastic strain $\varepsilon_m$ of a rock mass can be expressed as follows:

$$\varepsilon_m = \frac{\sigma_{cm}}{E} = \frac{\sigma_c}{E(1 - D_R)} = \frac{\sigma_c}{E_m}, \tag{3.3}$$

where $E_m$ is the elastic modulus of the rock mass, then

$$E_m \varepsilon_m = E_r \varepsilon_r. \tag{3.4}$$

## 3.2. Damage index and integrality coefficient

A rock mass is composed of statistically distributed joints/fracture and intact rock blocks. Hence, intact rock can be regarded as a homogeneous material, and a rock mass or damaged rock can be regarded as a heterogeneous material. Based on the theory of elastic waves, it is well known that the P-wave velocity $V_p$ in homogeneous material (intact rock) is expressed as follows:

$$V_p = \sqrt{\frac{E'_r(1 - \mu)}{\rho(1 + \mu)(1 - 2\mu)}}, \tag{3.5}$$

where $V_p$, $E'_r$, $\mu$ and $\rho$ are the P-wave velocity, dynamic modulus of elasticity, Poisson's ratio and density of the intact rock, respectively. The P-wave velocity $V'_p$ in heterogeneous material (rock mass or damaged rock) is expressed as follows:

$$V'_p = \sqrt{\frac{E'_m(1 - \mu')}{\rho'(1 + \mu')(1 - 2\mu')}}, \tag{3.6}$$

where $V'_p$, $E'_m$, $\mu'$ and $\rho'$ are the P-wave velocity, dynamic modulus of elasticity, Poisson's ratio and density of the joint rock mass or damaged rock mass, respectively.

Palmstrom & Singh [43] proposed that the ratio of the static modulus of elasticity to the dynamic modulus of elasticity of intact rock is equal to that of a damaged rock or rock mass. The function is expressed as follows:

$$\frac{E_m}{E'_m} = \frac{E_r}{E'_r}. \tag{3.7}$$

In combination, equations (3.2) and (3.7) can be expressed as follows:

$$E'_m = E'_r(1 - D_R). \tag{3.8}$$

From the point of view of rock engineering, Poisson's ratio and density of a rock mass and those of a rock block are approximately the same. The properties are expressed as follows:

$$\rho' = \rho \tag{3.9}$$

and

$$\mu' = \mu. \tag{3.10}$$

Therefore, the new damage index $D_R$ can be expressed as follows:

$$D_R = 1 - \frac{V_p'^2}{V_p^2}. \tag{3.11}$$

Equation (3.11) can also be expressed as follows:

$$D_R + K_V = 1, \tag{3.12}$$

where $K_V$ is the integrality coefficient of the rock mass, as indicated in equation (2.3).

The results show that the integrality coefficient is equal to 1 and the damage index is 0 in intact rock. When the rock is absolutely destroyed, the integrality coefficient is 0, and the damage index is 1. In practical engineering, a rock mass is variably damaged under the effects of complex geologic and engineering activities; thus, practically no absolute intact rock occurs in nature.

# 4. Experimental scheme and results

To verify the new discovery discussed in §2 and the theoretical derivation presented in §3, experiments were performed and are detailed in this section; these include a field point load test and a P-wave velocity test.

## 4.1. Point load test results

To verify the results presented in table 5, 15 testing sites were prepared for measuring the field point load strength and P-wave velocity in each lithology of the strata. When the *in situ* point load test samples were collected, some blocks of irregular joints were also gathered deliberately. A total of 30 samples per test site were prepared to conduct point load strength tests; the test method is discussed in §2. The measured results were recorded clearly for the average values of PLS for intact rock failure and splitting along a pre-existing joint, as listed in table 6, where the letter 'I' represents samples with intact rock failure, the letter 'J' represents samples with splitting along a pre-existing joint, $n$ is the number of effective trials performed at each testing site and $N$ is the total number of samples at each testing site.

## 4.2. P-wave velocity test results

Ultrasonic methods are non-destructive and relatively easy to use, both in the field and in the laboratory. The P-wave velocity has been used to characterize rock masses by many researchers [44–49]. In this study, the P-wave velocity of rock masses and that of intact rock were measured to calculate the integrality coefficient of the rock masses. The P-wave velocity of the rock masses was determined in the field, while that of intact rock was performed in the laboratory.

The P-wave velocity of the rock masses was measured by a single-borehole testing method using an RSM-SY5(T) intelligent engineering instrument. The single-borehole P-wave velocity testing method is illustrated in figure 4.

As shown in figure 4, a launching transducer and two receiving transducers are placed in the borehole, with pure water used as the coupling medium. A P-wave is launched from transducer R and received by transducers R1 and R2. The P-wave propagates into the rock mass, through the water and along the wall of the borehole. The P-wave arrives at receiving transducer R1 at time $t_1$ and at receiving transducer R2 at time $t_2$. Therefore, the P-wave velocity between R1 and R2 (shown in figure 4) can be calculated as

$$\Delta t = t_2 - t_1 \tag{4.1}$$

and

$$V_p' = \frac{\Delta L}{\Delta t}, \tag{4.2}$$

where $V_p'$ is the P-wave velocity of the rock mass and $\Delta L$ is the P-wave propagation distance between receiving transducers R1 and R2.

**Table 6.** The average values of PLS on different rock types and different failure types.

| rock type | $C_{1d}$ | | | $C_{1b}$ | | | $C_{2w}$ | | | $P_{1q+m}$ | | |
|---|---|---|---|---|---|---|---|---|---|---|---|---|
| | n/N | I | J | n/N | I | J | n/N | I | J | n/N | I | J |
| Test 1 | 24/30 | 3.29 | 1.22 | 21/30 | 3.37 | 1.95 | 26/30 | 1.61 | 0.92 | 25/30 | 3.83 | 1.29 |
| Test 2 | 26/30 | 2.32 | 2.35 | 27/30 | 7.93 | 2.19 | 25/30 | 2.45 | 1.78 | 24/30 | 4.02 | 2.61 |
| Test 3 | 23/30 | 2.80 | 2.72 | 23/30 | 2.70 | 1.84 | 24/30 | 4.23 | 1.50 | 23/30 | 7.27 | 3.51 |
| Test 4 | 25/30 | 2.74 | 1.79 | 25/30 | 3.51 | 1.56 | 21/30 | 3.57 | 1.68 | 25/30 | 5.94 | 2.10 |
| Test 5 | 27/30 | 2.69 | 1.44 | 24/30 | 3.23 | 1.78 | 22/30 | 4.78 | 2.43 | 26/30 | 3.67 | 2.71 |
| Test 6 | 22/30 | 3.84 | 3.39 | 22/30 | 2.77 | 1.41 | 25/30 | 3.02 | 0.89 | 23/30 | 4.82 | 1.92 |
| Test 7 | 25/30 | 2.98 | 2.63 | 25/30 | 3.19 | 0.98 | 24/30 | 2.41 | 1.69 | 27/30 | 2.67 | 0.81 |
| Test 8 | 26/30 | 4.70 | 3.2 | 26/30 | 5.11 | 2.00 | 25/30 | 1.21 | 0.54 | 22/30 | 8.90 | 3.87 |
| Test 9 | 24/30 | 3.86 | 2.62 | 23/30 | 5.03 | 1.91 | 27/30 | 2.45 | 0.97 | 23/30 | 6.51 | 2.69 |
| Test 10 | 21/30 | 6.90 | 1.25 | 25/30 | 5.78 | 2.40 | 24/30 | 2.02 | 0.91 | 26/30 | 2.29 | 1.34 |
| Test 11 | 27/30 | 4.48 | 1.48 | 27/30 | 2.88 | 1.23 | 26/30 | 3.72 | 2.05 | 25/30 | 4.71 | 2.57 |
| Test 12 | 23/30 | 4.37 | 2.90 | 25/30 | 3.36 | 1.77 | 23/30 | 3.65 | 2.31 | 22/30 | 3.31 | 1.63 |
| Test 13 | 26/30 | 5.50 | 2.72 | 22/30 | 3.48 | 1.82 | 23/30 | 1.71 | 0.65 | 25/30 | 5.24 | 3.01 |
| Test 14 | 25/30 | 2.57 | 1.65 | 23/30 | 4.09 | 1.73 | 25/30 | 2.97 | 0.94 | 24/30 | 4.28 | 1.15 |
| Test 15 | 22/30 | 3.32 | 1.98 | 26/30 | 3.77 | 2.74 | 27/30 | 2.78 | 1.42 | 21/30 | 3.59 | 2.43 |

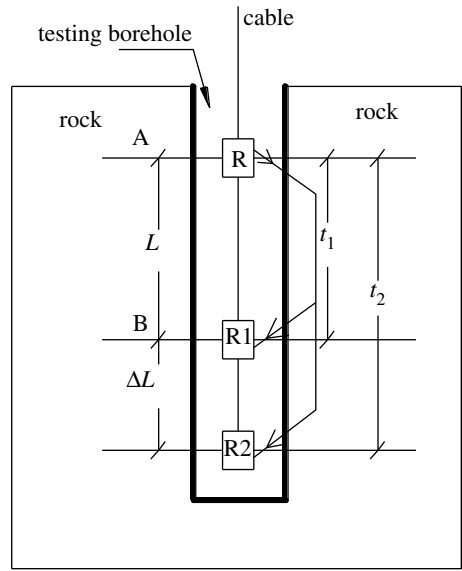

**Figure 4.** Single-borehole method for measuring P-wave velocity.

**Table 7.** The average values of P-wave velocity in rock mass and intact rock samples.

| rock type | $C_{1d}$ | | $C_{1b}$ | | $C_{2w}$ | | $P_{1q+m}$ | |
| | $N_1$ | P-velocity (m s$^{-1}$) | $N_1$ | P-velocity (m s$^{-1}$) | $N_1$ | P-velocity (m s$^{-1}$) | $N_1$ | P-velocity (m s$^{-1}$) |
|---|---|---|---|---|---|---|---|---|
| Test 1 | 5 | 4243.31 | 5 | 3284.80 | 5 | 3146.81 | 5 | 4376.60 |
| Test 2 | 5 | 3562.66 | 5 | 4261.30 | 5 | 2598.40 | 5 | 3354.04 |
| Test 3 | 5 | 3520.99 | 5 | 2991.28 | 5 | 3765.43 | 5 | 4014.25 |
| Test 4 | 5 | 3121.02 | 5 | 3906.89 | 5 | 3518.24 | 5 | 4341.73 |
| Test 5 | 5 | 3724.68 | 5 | 3441.31 | 5 | 3354.51 | 5 | 2917.73 |
| Test 6 | 5 | 3955.28 | 5 | 3554.17 | 5 | 4053.28 | 5 | 4306.57 |
| Test 7 | 5 | 3121.02 | 5 | 4199.09 | 5 | 2683.61 | 5 | 4546.96 |
| Test 8 | 5 | 3024.97 | 5 | 3973.68 | 5 | 3146.81 | 5 | 4089.29 |
| Test 9 | 5 | 3214.21 | 5 | 4071.81 | 5 | 3612.92 | 5 | 4126.30 |
| Test 10 | 5 | 4577.93 | 5 | 3873.07 | 5 | 3643.94 | 5 | 3573.48 |
| Test 11 | 5 | 4346.40 | 5 | 3906.89 | 5 | 3074.46 | 5 | 3615.77 |
| Test 12 | 5 | 3073.37 | 5 | 3479.34 | 5 | 2846.40 | 5 | 3898.99 |
| Test 13 | 5 | 3841.71 | 5 | 3591.00 | 5 | 3795.20 | 5 | 3657.57 |
| Test 14 | 5 | 3349.13 | 5 | 3940.43 | 5 | 3969.12 | 5 | 4580.27 |
| Test 15 | 5 | 3955.28 | 5 | 2856.26 | 5 | 3286.74 | 5 | 3262.13 |
| intact rock P-wave velocity | 5344 m s$^{-1}$ | | 5130 m s$^{-1}$ | | 4744 m s$^{-1}$ | | 5514 m s$^{-1}$ | |

The P-wave velocity of each rock mass was measured five times for each measuring point. The mean P-wave velocities are listed in table 7, where $N_1$ represents the number of times each sample was tested. The P-wave velocity of each rock mass was found to occupy a very broad range (2598.40−4580.27 m s$^{-1}$), varying widely between different types of rock mass and different test points. For the intact rock tested in the laboratory, the P-wave velocity varied from 3514 m s$^{-1}$ to 6075 m s$^{-1}$, and its mean value was found to occupy a very broad range (4744−5514 m s$^{-1}$).

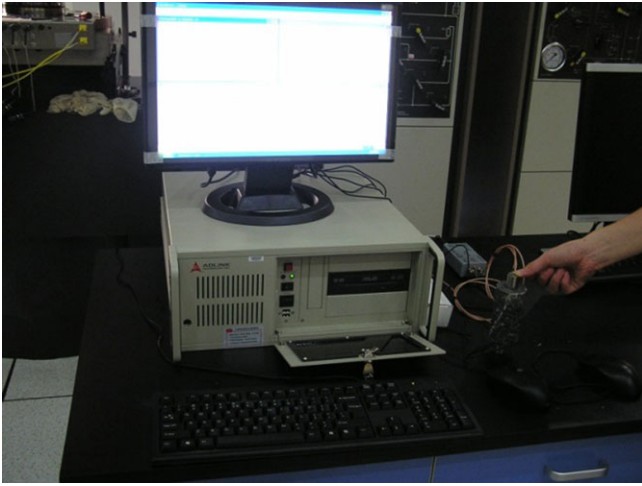

**Figure 5.** P-wave velocity test of intact rock.

The P-wave velocity of intact rock was measured using the ADLINK acoustic emission test system, which is made in the USA, as shown in figure 5. The formula for determining the P-wave velocity is as follows:

$$V_p = \frac{D}{t_p - t_0},$$

(4.3)

where $D$ is the centre distance between the launching transducer and receiving transducer, $t_p$ is the propagation time of the P-wave velocity in a rock sample, and $t_0$ is the zero delay of the instrument system. The mean P-wave velocities of different types of intact rock ($C_{1d}$, $C_{1b}$, $C_{2w}$ and $P_{1q+m}$) were determined to be 5344 m s$^{-1}$, 5130 m s$^{-1}$, 4744 m s$^{-1}$ and 5514 m s$^{-1}$, respectively.

# 5. The analysis of damage index

Determining how to quantify the degree of damage undergone by rock materials that are damaged but not destroyed has been the focus of many studies. As is well known, the strength and P-wave velocity of damaged rock are less than those of intact rock. In this regard, a new method for calculating the damage index $D_R$ was proposed based on the field point load strength test, as expressed in equation (2.4). The damage index $D_R$ can also be expressed in terms of the integrality coefficient $K_V$, as shown in equation (3.11). The integrality coefficient $K_V$ can be defined by the P-wave velocity, as shown in equation (2.3). To verify the accuracy and validity of the new damage index $D_R$, the point load strength and P-wave velocity were measured in the field.

The results are listed in table 8. All deviations are less than 10%, except for the one deviation of 11.87%. Moreover, the theoretical correlation between $D_R$ and $K_V$ satisfies the linear relationship established by equation (3.12). Thus, linear regression analysis was used to determine the relationship between $D_R$ and $K_V$, with the confidence limits set to 95%. The purpose of linear regression analysis is to validate the accuracy between a theoretical correlation (in this case, the correlation between $D_R$ and $K_V$) and test results. The linear regression equation takes the form $y = a + bx$, where $b$ is the regression coefficient. The parameter $a$ is a constant representing the value of $y$ when the independent variable $x$ is zero. The cross-correlations of the new damage index $D_R$ and the integrality coefficient $K_V$ for linear regression are shown graphically in figures 6–9. The black solid line is the theoretical line between $D_R$ and $K_V$ according to equation (3.12), the fitting line is the red solid line, and the confidence intervals are indicated by the blue dotted line.

All of the results demonstrate a reasonable linear relationship with a high coefficient of determination. The correlation coefficients $R^2$ of rock masses $C_{1d}$, $C_{1b}$, $C_{2w}$ and $P_{1q+m}$ are 0.98, 0.98, 0.97 and 0.99, respectively. Moreover, when the fitting straight line is close and parallel to the theoretical line, the damage index computed using the current method is in good agreement with the damage index computed based on the P-wave velocity. Overall, the results indicate that the current

**Table 8.** The calculation results of $D_R$ (AVE—absolute value of relative error).

| rock type | $C_{1d}$ | | | | $C_{1b}$ | | | | $C_{2w}$ | | | | $P_{1q+m}$ | | | |
|---|---|---|---|---|---|---|---|---|---|---|---|---|---|---|---|---|
| | $D_R$ | $1-D_R$ | $K_V$ | AVE | $D_R$ | $1-D_R$ | $K_V$ | AVE | $D_R$ | $1-D_R$ | $K_V$ | AVE | $D_R$ | $1-D_R$ | $K_V$ | AVE |
| Test 1 | 0.37 | 0.63 | 0.61 | 3.14% | 0.58 | 0.42 | 0.41 | 2.77% | 0.57 | 0.43 | 0.44 | 2.74% | 0.34 | 0.66 | 0.63 | 5.27% |
| Test 2 | 0.58 | 0.42 | 0.43 | 2.77% | 0.28 | 0.72 | 0.69 | 4.90% | 0.73 | 0.27 | 0.30 | 8.84% | 0.65 | 0.35 | 0.37 | 5.20% |
| Test 3 | 0.61 | 0.39 | 0.42 | 8.16% | 0.68 | 0.32 | 0.34 | 6.32% | 0.35 | 0.65 | 0.63 | 2.44% | 0.48 | 0.52 | 0.53 | 2.42% |
| Test 4 | 0.65 | 0.35 | 0.33 | 5.07% | 0.44 | 0.56 | 0.58 | 4.21% | 0.47 | 0.53 | 0.55 | 3.74% | 0.35 | 0.65 | 0.62 | 4.27% |
| Test 5 | 0.54 | 0.46 | 0.49 | 5.17% | 0.55 | 0.45 | 0.45 | 0 | 0.51 | 0.49 | 0.50 | 1.67% | 0.74 | 0.26 | 0.28 | 6.58% |
| Test 6 | 0.49 | 0.51 | 0.53 | 4.19% | 0.51 | 0.49 | 0.48 | 2.29% | 0.29 | 0.71 | 0.73 | 3.38% | 0.40 | 0.60 | 0.61 | 1.37% |
| Test 7 | 0.68 | 0.32 | 0.33 | 3.40% | 0.31 | 0.69 | 0.67 | 3.40% | 0.70 | 0.30 | 0.32 | 6.64% | 0.30 | 0.70 | 0.68 | 2.45% |
| Test 8 | 0.71 | 0.29 | 0.31 | 5.28% | 0.39 | 0.61 | 0.60 | 1.44% | 0.45 | 0.55 | 0.44 | 2.54% | 0.43 | 0.57 | 0.55 | 2.76% |
| Test 9 | 0.68 | 0.32 | 0.35 | 8.22% | 0.38 | 0.62 | 0.63 | 1.54% | 0.40 | 0.60 | 0.58 | 4.15% | 0.41 | 0.59 | 0.56 | 4.78% |
| Test 10 | 0.30 | 0.70 | 0.73 | 3.71% | 0.42 | 0.58 | 0.57 | 2.59% | 0.45 | 0.55 | 0.59 | 6.86% | 0.59 | 0.41 | 0.42 | 1.23% |
| Test 11 | 0.33 | 0.67 | 0.64 | 4.63% | 0.43 | 0.57 | 0.58 | 1.22% | 0.55 | 0.45 | 0.42 | 6.89% | 0.55 | 0.45 | 0.43 | 5.66% |
| Test 12 | 0.66 | 0.34 | 0.32 | 5.12% | 0.53 | 0.47 | 0.46 | 2.87% | 0.63 | 0.37 | 0.36 | 1.98% | 0.49 | 0.51 | 0.50 | 1.51% |
| Test 13 | 0.49 | 0.51 | 0.50 | 1.09% | 0.52 | 0.48 | 0.49 | 2.65% | 0.38 | 0.62 | 0.64 | 3.14% | 0.57 | 0.43 | 0.44 | 3.28% |
| Test 14 | 0.64 | 0.36 | 0.38 | 5.80% | 0.42 | 0.58 | 0.59 | 2.20% | 0.32 | 0.68 | 0.70 | 2.36% | 0.27 | 0.73 | 0.69 | 5.99% |
| Test 15 | 0.45 | 0.55 | 0.53 | 4.57% | 0.73 | 0.27 | 0.31 | 11.87% | 0.51 | 0.49 | 0.48 | 1.92% | 0.68 | 0.32 | 0.35 | 7.68% |

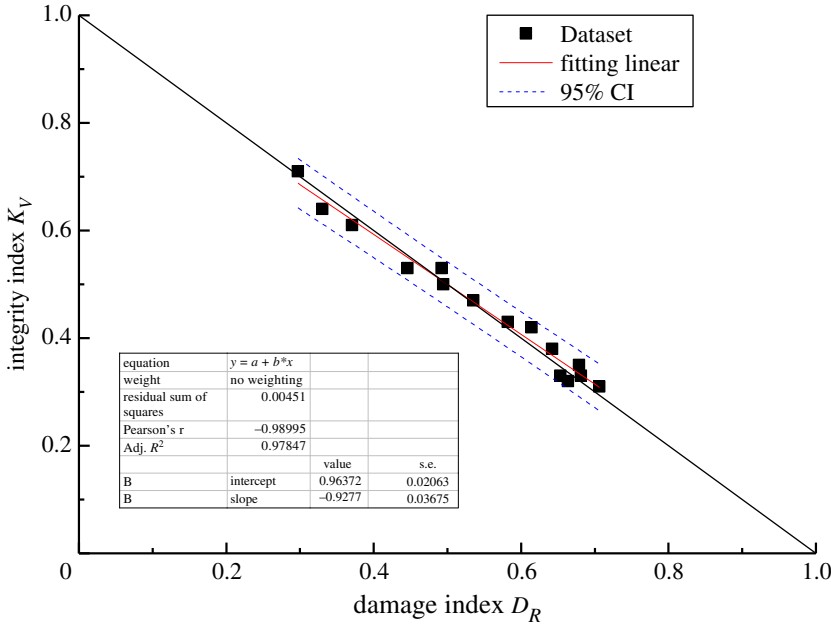

**Figure 6.** Comparison of the $D_R$ value determined by the new method with the theoretical value for $C_{1d}$.

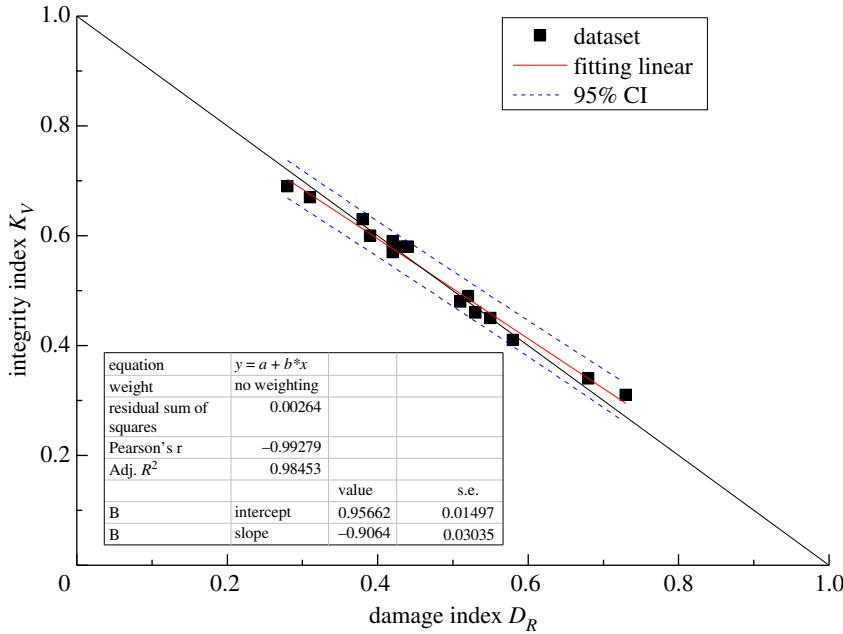

**Figure 7.** Comparison of the $D_R$ value determined by the new method with the theoretical value for $C_{1b}$.

method is effective and can be used to calculate the elastic modulus and unconfined compressive strength of rock masses.

## 6. Discussion

Two failure modes, intact rock failure and splitting along a pre-existing joint, were observed in the field point load strength experiment. The new damage index $D_R$ is defined as a point load strength ratio whose numerator is the strength of splitting along a pre-existing joint and whose denominator is the strength of intact rock failure. The strength ratio is very similar to the damage index of rock masses.

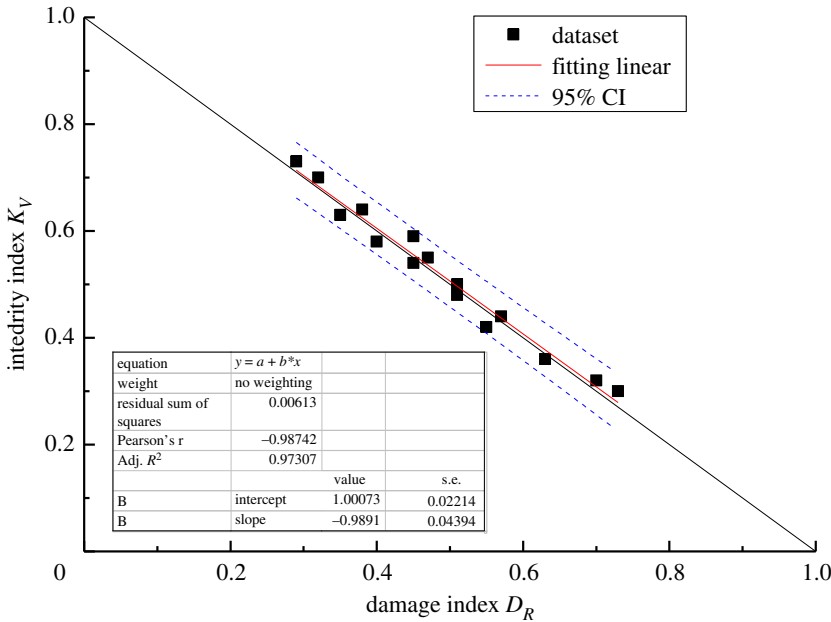

**Figure 8.** Comparison of the $D_R$ value determined by the new method with the theoretical value for $C_{2w}$.

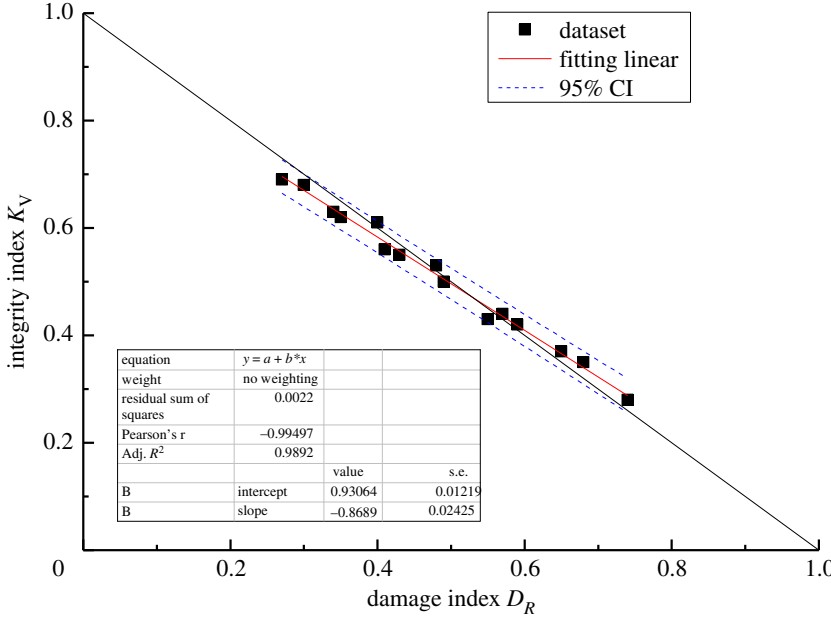

**Figure 9.** Comparison of the $D_R$ value determined by the new method with the theoretical value for $P_{1q+m}$.

Hence, *in situ* tests were designed to verify the newly developed index $D_R$, including an acoustic emission test, a P-wave test and a field point load strength test.

Theoretical derivations indicate that it is feasible to use the point load strength to define the damage index of a rock mass. The theoretical results obtained in this study show that the ratio between the point load strength of splitting along a pre-existing joint and that of intact rock failure is correlated with the integrity coefficient. Typically, the integrity coefficient of a rock mass decreases with increasing joint density, as occurs in the formation of micro-cracks or macroscopic fractures. Hence, the degree of damage sustained by a rock mass must be assessed in the study area. This study considered the point load strength of rock masses undergoing splitting along a pre-existing joint or intact rock failure. However, the degree to which joints developed in the samples and the direction along which they did so could not be accurately measured. We assumed that the splitting of a sample along a pre-existing

joint always occurs along the weakest surface. Thus, the newly developed $D_R$ is determined solely by the failure property of field point load strength.

# 7. Conclusion

Based on the statistical results obtained from field point load strength tests, two different failure types were discovered. The specific relationship between the field point load strength ratio $D_R$ and the integrity coefficient $K_V$ of rock masses was determined in this study. To this end, theoretical derivations (§3) and a series of tests (§4) were conducted. Finally, a new method for calculating the damage index in terms of the field point load strength was proposed. Compared with current methods for determining the integrality coefficient, which is the ratio between the P-wave velocity measured in the laboratory and that measured in the field, this new method provides an easier and faster way of estimating the integrality coefficient or the damage index of rock masses. Moreover, the method can be used to estimate the elastic modulus and unconfined compressive strength of rock masses. Therefore, the technique has broad application prospects in underground geotechnical engineering. In conclusion, this new method for calculating the damage index should be further tested and verified by conducting more empirical strength tests, such as those measuring uniaxial compressive strength, uniaxial tensile strength and triaxial compressive strength.

Ethics. This work having obtained permission from all the authors, we declare that: (a) the material has not been published in whole or in part elsewhere; (b) the paper is not currently being considered for publication elsewhere; (c) all authors have been personally and actively involved in substantive work leading to the report, and will hold themselves jointly and individually responsible for its content; (d) all relevant ethical safeguards have been met in relation to patient or subject protection, or animal experimentation.

Data accessibility. Our data are deposited at Dryad Digital Repository: http://dx.doi.org/10.5061/dryad.bm76m89 [50].

Authors' contributions. L.W. and Z.q.L. designed the study. S.j.Y. and L.W. prepared point load strength samples for analysis. S.w.M. and H.J. prepared P-wave velocity samples for analysis. L.W. and S.j.Y. collected and analysed the data. L.W. and Y.g.Q. interpreted the results and wrote the manuscript. All authors gave final approval for publication.

Competing interests. We declare we have no competing interests.

Funding. Financial support came from the National Key R&D Program of China during the 13th Five Year Plan Period: The Continuous Mining Theory and Technology on Spatiotemporal Synergism of Multi-mining Areas within a Large Ore Block for Deep Metal Deposit (grant no. 2017YFC0602901). And The Graduate Students Explore Innovative Projects Independently program (grant no. 2017zzts188).

Acknowledgements. We thank instructional support specialist Modern Analysis and Testing Central of Central South University. We are also grateful to my teachers and colleagues for their contributions to the article. We are also grateful to the reviewers and the editors for their comments and suggestions, which helped improve the manuscript.

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
