## [Reviewer comments · Royal Society Open Science]

Review History

RSOS-181591.R0 (Original submission)

Review form: Reviewer 1 (Cheng yu Xie)

Is the manuscript scientifically sound in its present form?

Yes

Are the interpretations and conclusions justified by the results?

Yes

Is the language acceptable?

Yes

Is it clear how to access all supporting data?

Yes

Do you have any ethical concerns with this paper?

No

Have you any concerns about statistical analyses in this paper?

No

Recommendation?

Accept with minor revision (please list in comments)

Comments to the Author(s)

Evaluation of the engineering properties of rock mass (damage index and integrity coefficient) is quite interesting topics in the scopes of rock mechanics and rock engineering. The field point load test was considered as a cheap and useful testing method due to its ease of testing, simplicity of specimen preparation, and widespread field application. In this manuscript, a new method was proposed to evaluate the engineering properties of rock mass by discussing field point load testing results. A set of tests and some statistical analyses have been carried out in this manuscript, which can be provided some potential value for further researches or engineering practices. The content of the paper does relay some interesting findings and therefore can be published if the minor remarks are corrected. Some minor remarks for quality improvement of this manuscript are shown as follows:

1. Page 1 Line 44-46: The sentence 'Several researchers (Aydan O et al.,...) reported the σ_{cm}/σ_c versus'. What does mean the ' σ_{cm}/σ_c ', it represents 'the ratio' or ' σ_{cm} or σ_c '?
2. Abbreviations should be used in uniformity, such in the Eq. 2 and Eq. 4. In the Eq. 2, "PLS" can be replaced by "Is". If want to agreed with the conventional usage, I suggest use "PLS" for the point load strength in the Eq. 2.
3. A notation can be added after Table 3, to give the definitions for all variables and explain the parameters L, W, H, I, J, HD and Wf, although these definitions have been appeared in the paper.
4. Different notations are listed in Table 3 and Table 4. In the Table 4, what does the 'P' present?
5. Page 9 Line 31-32: What does mean the Eq.8?
6. In the Table 7, the 'Velocity' should be replaced by 'P-wave velocity'.

Review form: Reviewer 2 (Ashutosh Trivedi)**Is the manuscript scientifically sound in its present form?**

Yes

Are the interpretations and conclusions justified by the results?

Yes

Is the language acceptable?

Yes

Is it clear how to access all supporting data?

Yes

Do you have any ethical concerns with this paper?

No

Have you any concerns about statistical analyses in this paper?

Yes

Recommendation?

Major revision is needed (please make suggestions in comments)

Comments to the Author(s)

This paper considers a damage index of rock masses which is a crucial controlling parameter for rock mass strength and deformation in civil, geological and mining engineering projects. In this study, a method was proposed for computing the damage index of a rock mass based on the field point load strength. This method considers a strength ratio whose numerator is the point load strength (PLS) of splitting along the pre-existing joint and whose denominator is the PLS of intact rock failure. To validate this method, the authors derived a relationship between the damage index and the integrality coefficient using an empirical relation. Moreover, numerous experimental tests were conducted, including P-wave testing and on-site point load testing. Finally, linear regression analysis was performed to analyze the correlation between the new damage index R_D and the integrality coefficient v_K . The authors claim that they have presented new method which is sensitive to both plasticity and damage in estimating the damage degree of rock masses in underground excavation and tunnel engineering.

The claim that the model takes care of plasticity is unfounded.

It is not clear how the wave velocity which is the mass response reflected in the integrality index has linear regression with the PLS ratio which is just surface response.

The authors should improve the m/s based upon the comments made on the paper.

Decision letter (RSOS-181591.R0)

25-Jan-2019

Dear Dr Wen

On behalf of the Editors, I am pleased to inform you that your Manuscript RSOS-181591 entitled "A New Method for Evaluating the Rock Mass Damage Index Based on the Field Point Load Strength" has been accepted for publication in Royal Society Open Science subject to minor revision in accordance with the referee suggestions. Please find the referees' comments at the end of this email.

The reviewers and handling editors have recommended publication, but also suggest some minor revisions to your manuscript. Therefore, I invite you to respond to the comments and revise your manuscript.

- Ethics statement

- Data accessibility

It is a condition of publication that all supporting data are made available either as supplementary information or preferably in a suitable permanent repository. The data accessibility section should state where the article's supporting data can be accessed. This section should also include details, where possible of where to access other relevant research materials

such as statistical tools, protocols, software etc can be accessed. If the data has been deposited in an external repository this section should list the database, accession number and link to the DOI for all data from the article that has been made publicly available. Data sets that have been deposited in an external repository and have a DOI should also be appropriately cited in the manuscript and included in the reference list.

If you wish to submit your supporting data or code to Dryad (<http://datadryad.org/>), or modify your current submission to dryad, please use the following link:
<http://datadryad.org/submit?journalID=RSOS&manu=RSOS-181591>

- **Competing interests**

- **Authors' contributions**

- **Acknowledgements**

- **Funding statement**

Because the schedule for publication is very tight, it is a condition of publication that you submit the revised version of your manuscript before 03-Feb-2019. Please note that the revision deadline will expire at 00.00am on this date. If you do not think you will be able to meet this date please let me know immediately.

To revise your manuscript, log into <https://mc.manuscriptcentral.com/rsos> and enter your Author Centre, where you will find your manuscript title listed under "Manuscripts with Decisions". Under "Actions," click on "Create a Revision." You will be unable to make your

revisions on the originally submitted version of the manuscript. Instead, revise your manuscript and upload a new version through your Author Centre.

Once again, thank you for submitting your manuscript to Royal Society Open Science and I look

forward to receiving your revision. If you have any questions at all, please do not hesitate to get in touch.

Kind regards,
 Andrew Dunn
 Senior Publishing Editor
 Royal Society Open Science
 openscience@royalsociety.org

on behalf of Dr Ian Moore (Associate Editor) and R. Kerry Rowe (Subject Editor)
 openscience@royalsociety.org

Associate Editor Comments to Author (Dr Ian Moore):

The second reviewer indicates that one claim from the work is unfounded (related to plasticity effects), but otherwise both reviewers seem positive about the work and have suggested various minor revisions. Please undertake minor revisions to address each of the reviewer suggestions and concerns, and resubmit for the final decision.

Associate Editor: 2

Comments to the Author:

I judge the work to be novel and well written and I am recommending it for review.

Reviewer comments to Author:

Reviewer: 1

Comments to the Author(s)

Evaluation of the engineering properties of rock mass (damage index and integrity coefficient) is quite interesting topics in the scopes of rock mechanics and rock engineering. The field point load test was considered as a cheap and useful testing method due to its ease of testing, simplicity of specimen preparation, and widespread field application. In this manuscript, a new method was proposed to evaluate the engineering properties of rock mass by discussing field point load testing results. A set of tests and some statistical analyses have been carried out in this manuscript, which can be provided some potential value for further researches or engineering practices. The content of the paper does relay some interesting findings and therefore can be published if the minor remarks are corrected. Some minor remarks for quality improvement of this manuscript are shown as follows:

1. Page 1 Line 44-46: The sentence 'Several researchers (Aydan O et al.,...) reported the σ_{cm}/σ_c versus'. What does mean the ' σ_{cm}/σ_c ', it represents 'the ratio' or ' σ_{cm} or ' σ_c '?
2. Abbreviations should be used in uniformity, such in the Eq. 2 and Eq. 4. In the Eq. 2, "PLS" can be replaced by "Is". If want to agreed with the conventional usage, I suggest use "PLS" for the point load strength in the Eq. 2.
3. A notation can be added after Table 3, to give the definitions for all variables and explain the parameters L, W, H, I, J, HD and Wf, although these definitions have been appeared in the paper.
4. Different notations are listed in Table 3 and Table 4. In the Table 4, what does the 'P' present?
5. Page 9 Line 31-32: What does mean the Eq.8?
6. In the Table 7, the 'Velocity' should be replaced by 'P-wave velocity'.

Reviewer: 2

Comments to the Author(s)

This paper considers a damage index of rock masses which is a crucial controlling parameter for rock mass strength and deformation in civil, geological and mining engineering projects. In this study, a method was proposed for computing the damage index of a rock mass based on the field point load strength. This method considers a strength ratio whose numerator is the point load strength (PLS) of splitting along the pre-existing joint and whose denominator is the PLS of intact rock failure. To validate this method, the authors derived a relationship between the damage index and the integrality coefficient using an empirical relation. Moreover, numerous experimental tests were conducted, including P-wave testing and on-site point load testing. Finally, linear regression analysis was performed to analyze the correlation between the new damage index R_D and the integrality coefficient v_K . The authors claim that they have presented new method which is sensitive to both plasticity and damage in estimating the damage degree of rock masses in underground excavation and tunnel engineering.

The claim that the model takes care of plasticity is unfounded.

It is not clear how the wave velocity which is the mass response reflected in the integrality index has linear regression with the PLS ratio which is just surface response.

The authors should improve the m/s based upon the comments made on the paper.

Author's Response to Decision Letter for (RSOS-181591.R0)

See Appendix A.

Decision letter (RSOS-181591.R1)

05-Feb-2019

Dear Dr Wen,

I am pleased to inform you that your manuscript entitled "A New Method for Evaluating the Rock Mass Damage Index Based on the Field Point Load Strength" is now accepted for publication in Royal Society Open Science.

Royal Society Open Science operates under a continuous publication model

(<http://bit.ly/cpFAQ>). Your article will be published straight into the next open issue and this will be the final version of the paper. As such, it can be cited immediately by other researchers. As the issue version of your paper will be the only version to be published I would advise you to check your proofs thoroughly as changes cannot be made once the paper is published.

on behalf of Dr Ian Moore (Associate Editor) and Professor R. Kerry Rowe (Subject Editor)
openscience@royalsociety.org

Appendix A

List of Responses

Dear Editors and Reviewers:

Thank you for your letter and for the reviewer's comments concerning our manuscript entitled "A New Method for Evaluating the Rock Mass Damage Index Based on the Field Point Load Strength" (ID: RSOS-181591). Those comments are all valuable and very helpful for revising and improving our paper, as well as the important guiding significance to our researches. We have studied comments carefully and have made corrections which we hope meet with approval. Revised or added portions are marked in red in the paper. Many explanations are marked in the form of annotations. The main corrections in the paper and the responses to the reviewer's comments are as follows:

Reviewer: 1

Response to the reviewer's comments:

1. Response to comment: Page 1 Line 44-46: The sentence 'Several researchers (Aydan O et al.,...) reported the σ_{cm}/σ_c versus'. What does mean the ' σ_{cm}/σ_c ', it represents 'the ratio' or ' σ_{cm} or σ_c '?

Response: We are very sorry for our negligence of the detail. In this paper, the ' σ_{cm}/σ_c ' represents 'the ratio'.

2. Response to comment: Abbreviations should be used in uniformity, such in the Eq. 2 and Eq. 4. In the Eq. 2, " PLS" can be replaced by " Is". If want to agreed with the conventional usage, I suggest use " PLS" for the point load strength in the Eq. 2.

Response: It is very sorry that we ignored the abbreviations uniformity. We have corrected according with reviewer's suggestion.

3. Response to comment: A notation can be added after Table 3, to give the definitions for all variables and explain the parameters L, W, H, I, J, HD and Wf, although these definitions have been appeared in the paper.

Response: It is very sorry that we ignored the notation after Table 3. A notation has been added after Table 3, to give the definitions for all variables.

4. Response to comment: Different notations are listed in Table 3 and Table 4. In the Table 4, what does the 'P' present?

Response: We are very sorry for the inconsistency between the letters in Table 3 and Table 4. In this regard, we have made a detailed inspection and correction.

5. Response to comment: Page 9 Line 31-32: What does mean the Eq.8?

Response: We are very sorry for the detail of the Eq.8 is ignored by us. We have

corrected the correctness of Eq.8.

6. Response to comment: In the Table 7, the 'Velocity' should be replaced by 'P-wave velocity'.

Response: We are very sorry for the details are ignored by us. It has been corrected in the Table 7.

Reviewer: 1

Responds to the reviewer's comments:

1. Response to comment: The claim that the model takes care of plasticity is unfounded.

Response: Rock mass is a brittle material. When cracks exist in rock mass, the plastic deformation of rock mass will be occurred under loading.

2. Response to comment: It is not clear how the wave velocity which is the mass response reflected in the integrality index has linear regression with the PLS ratio which is just surface response.

Response: The PLS ratio has relatively few studies. In the next study, the size, density and closure of joint surface will be further studied and discussed.